# Short-Term Outcomes of Surgery and Rehabilitation on Activities of Daily Living after Displaced Femoral Neck Fractures: Structural Equation Modeling

**DOI:** 10.3390/jcm12031234

**Published:** 2023-02-03

**Authors:** Kazutaka Yokoyama, Hiroyuki Katoh, Seiji Bito, Yoshinari Fujita, Keita Yamauchi

**Affiliations:** 1Graduate School of Health Management, Keio University, Fujisawa 252-0883, Japan; 2Department of Orthopedic Surgery, Tokai University School of Medicine, Isehara 259-1193, Japan; 3Department of Clinical Epidemiology, National Hospital Organization Tokyo Medical Center, Meguro 152-8902, Japan; 4Department of Orthopedic Surgery, National Hospital Organization Tokyo Medical Center, Meguro 152-8902, Japan

**Keywords:** bipolar hemiarthroplasty, hip fractures, multiple factors, multiple regression analysis, total hip arthroplasty

## Abstract

In order to explore the factors affecting patients’ level of activities of daily living (ADL) on discharge after undergoing bipolar hemiarthroplasty or total hip arthroplasty for displaced femoral neck fractures at an acute care hospital, patient data were analyzed with the following statistical tools: multiple regression analysis (MRA), structural equation modeling (SEM), and simultaneous analysis of several groups (SASG). The Barthel Index (BI) on discharge was set as the objective variable, while age, sex, degree of dementia, BI on admission, number of days from admission to surgery, surgical option, and number of rehabilitation units per day were set as explanatory variables. Factors such as age, sex, degree of dementia, BI on admission, and number of rehabilitation units per day were significant in MRA. While not significant in MRA, the number of days from admission to surgery was significant in SEM. According to the SASG, the number of rehabilitation units per day was significant for patients without dementia but not for patients with dementia. Analysis of real-world data suggests that early surgery and rehabilitation affect ADL on discharge to a greater degree than the surgical method. For patients without dementia, longer daily rehabilitation was significantly associated with better ADL on discharge.

## 1. Introduction

Hip fractures cause serious problems in aging populations worldwide. The number of patients with new hip fractures is increasing annually in Japan, with 76,600 cases reported in 1992, increasing to 193,400 cases in 2017 [1], and is forecasted to reach 290,000 cases in 2030 [2]. The poor prognosis after hip fractures significantly impacts the patient and their family and poses a great burden on society because a person who lives independently will often need nursing care or become bedridden after sustaining a hip fracture.

According to a systematic review reported in 2019, the predictors of functional outcome and mortality after hip fractures can be grouped into four groups: (1) medical factors such as cognitive impairment, (2) surgical factors such as delay in operation, (3) socioeconomic factors such as age and sex, and (4) other factors such as length of hospital stay [3]. In 2021, dementia was reported to significantly negatively impact postoperative walking ability [4]. In 2022, rehabilitation intensity was reported to affect activities of daily living (ADL) [5].

The effects of surgical factors on surgeons are an issue of interest. One factor that was analyzed was the delay in surgery. According to a survey of hip fractures in Japan, the mean waiting time before surgery for a femoral neck fracture was 4.9 days in 2014 [6]. Since early surgery is associated with fewer complications, higher survival rates, and shorter hospital stay, the *Japanese Orthopedic Association Clinical Practice Guideline on the Management of Hip Fractures*, 3rd Edition (2021), recommends surgery as early as possible [2,7,8,9,10,11], and the delay in surgery has been decreasing in Japan. Another factor of interest was the surgical method used by the surgeon. For displaced femoral neck fractures, surgeons choose between bipolar hemiarthroplasty (BHA) and total hip arthroplasty (THA), but the difference between these procedures is controversial. Blomfeldt et al. (2007) reported that THA provided better hip function than BHA without increasing the complication rate [12], but a systematic review (2010) found only limited evidence that THA led to better functional outcomes than BHA [13]. BHA produces stable results, but outcomes are suboptimal in cases with damaged acetabular cartilage or acetabular labral tear. Furthermore, considering that the pain caused by long-term degeneration of the acetabular cartilage after BHA may lead to reoperation, some surgeons recommend THA as the primary surgery. However, since THA carries a risk of dislocation, an anterior or anterolateral approach and/or the use of a dual-mobility cup may be recommended. Surgeons at the National Hospital Organization Tokyo Medical Center (NTMC) recommended BHA to treat dislocated femoral neck fractures, except for younger, highly demanding patients with a wide range of activity, in which case THA through an anterolateral muscle-sparing approach to reduce the risk of dislocation was recommended.

The Diagnosis Procedure Combination (DPC) system is a unique Japanese system introduced in April 2003 to classify patients admitted in the acute phase of illness. It was originally developed as a method to standardize medical care and evaluate its quality but has since been incorporated into the Japanese medical care reimbursement system to allocate appropriate medical resources [14]. In this study, we used the following information recorded in the DPC data: patient demographics, type of surgery, ADL scores, and dates related to hospitalization. The ADL scores, with a maximum score of 20, were converted into Barthel Index (BI) scores, which are cumulative scores of 10 items with varying weights of ADL points. A maximum BI score of 100 denotes complete independence, and a minimum score of 0 corresponds to total dependence [15]. The absence or degree of dementia and the number of rehabilitation units were extracted from the hospital’s electronic medical records. The number of rehabilitation units refers to the duration of rehabilitation training, counted as one unit when patients were provided with 20 min of individual therapy.

By analyzing information extracted from DPC data and electronic medical records, this study aimed to explore how multiple factors, such as surgery and rehabilitation, affect final ADL on discharge. In addition to the traditional multiple regression analysis (MRA), structural equation modeling (SEM) and simultaneous analysis of several groups (SASG) of SEM were employed. SEM is a powerful statistical technique that allows the testing of complex relationships between variables specified within a hypothesized model [16], making it possible to visualize the relationships among variables.

## 2. Materials and Methods

### 2.1. Study Design

We performed a retrospective observational study using patient-level data extracted from the DPC data and the electronic medical record system stored and maintained by the NTMC.

DPC data from patients who were admitted to the NTMC for a displaced femoral neck fracture and underwent BHA or THA between April 2015 and March 2021 were collected, excluding reoperation cases and patients treated while under the care of other departments.

### 2.2. Variables

Patient demographics (e.g., age and sex), surgical options (e.g., BHA and THA), ADL scores (e.g., BI on admission and BI on discharge), and hospitalization-related dates (e.g., dates of admission, surgery, and discharge) were collected from the DPC data. Days from admission to surgery and duration of hospitalization were calculated from hospitalization-related dates. Data on the degree of dementia, number of rehabilitation units, and rehabilitation duration were extracted from the electronic medical record system. The degree of dementia was classified as severe (requiring nursing care), moderate (independent), or no cognitive impairment. The number of rehabilitation units per day was calculated by dividing the number of rehabilitation units by the number of days of rehabilitation.

As the ADL score on discharge was the primary outcome of this study, we set the objective variable as BI on discharge (Y) and the explanatory variables as follows: age (X1), sex (X2), degree of dementia (X3), BI on admission (X4), days from admission to surgery (X5), surgical option (X6), and number of rehabilitation units per day (X7).

### 2.3. Statistical Analysis

All data were collected and organized using Microsoft Excel (Microsoft Corporation, Richmond, WA, USA). Patient data were anonymized before analysis. Student’s *t*-test was used for continuous variables, and Fisher’s exact test was used for categorical variables when patients’ demographic characteristics were summarized. For MRA, standard error (SE), variance inflation factor (VIF), R-squared (R^2^), adjusted R-squared (adjusted R^2^), and root mean square error (RMSE) were calculated using JMP 16 for Mac (SAS Institute Inc., Cary, NS, USA). SEM and SASG were performed using JUSE-StatWorks/V5 (Version 5.82, The Institute of Japanese Union of Scientists & Engineers, Tokyo, Japan), and the fit of the model was assessed using the comparative fit index (CFI), goodness-of-fit index (GFI), adjusted goodness-of-fit index (AGFI), and root mean square error of approximation (RMSEA). The goodness-of-fit was evaluated according to the following cut-off criteria: CFI ≥ 0.95, GFI ≥ 0.95, AGFI ≥ 0.90, and RMSEA ≤ 0.07 [17]. For the SASG, an equality constraint was added to the arrow from X5 to Y in both groups to account for the fact that the days from admission to surgery would affect all patients equally.

## 3. Results

A total of 283 patients underwent surgery between April 2015 and March 2021, of which 15 and 7 cases were excluded because of reoperations and hospitalizations in other departments, respectively. Finally, 261 cases consisting of 223 BHAs and 38 THAs were included in this study. Patients’ demographic characteristics, grouped according to the surgical method, are summarized in Table 1. Between the two groups, the mean length of hospital stay was similar (29.5 days vs. 26.7 days), but the mean number of days from admission to surgery was significantly shorter in the BHA group compared to that of the THA group (4.0 days vs. 6.9 days). The characteristics evident from this comparison reflect the surgical indications for THA in NTMC: patients in the BHA group were significantly older (84.5 years vs. 66.3 years), more likely to have dementia (53.8% vs. 13.2%), and had a lower BI on admission (15.6 points vs. 30.5 points) than patients in the THA group.

Since dementia has been demonstrated to be a crucial risk factor for poor prognosis after hip fracture surgery [4], we analyzed its effect on changes in ADL status (Figure 1). BI scores on admission and discharge were plotted, with a red line indicating severe dementia, a blue line indicating no cognitive impairment, and a yellow line indicating moderate impairment. The overlapping of multiple lines accounts for the different hues and the darker color. The results of this analysis demonstrated that a greater improvement in ADL was observed in patients without dementia.

MRA revealed that age, sex, degree of dementia, BI on admission, and number of rehabilitation units per day were significant factors affecting ADL on discharge (Table 2). According to MRA, the formula for BI on discharge according to the explanatory variables can be presented as follows:Y = −0.51X1 + 9.28X2 − 23.83X3 + 0.36X4 − 0.60X5 + 2.49X6 + 18.71X7 + 61.37

VIFs were calculated to verify the independence of the explanatory values, revealing a high VIF of 2.37 in the age variable. Correlation coefficients calculated to evaluate multicollinearity among the variables (Table 3) demonstrated a significant correlation between age and the following factors: degree of dementia, BI on admission, days from admission to surgery, surgical option, and number of rehabilitation units per day. Therefore, SEM was performed to solve multicollinearity, taking into consideration the factors with significant correlations (*p* < 0.01) (Figure 2). Table 4 shows the direct effect, indirect effect, total effect, and P-value of the SEM, which are reflected as either one- or two-direction arrows in Figure 2 to convey causality or correlation among factors, respectively. The number of days from admission to surgery was not statistically significant in the MRA (*p* = 0.056) but was significant in the SEM images (*p* = 0.049). This discrepancy was further analyzed using SASG, and the results are presented in Figure 3. The SASG demonstrated that the number of days from admission to surgery was a significant contributing factor to ADL on discharge in both groups. As for factors affected by the medical staff, the number of rehabilitation units per day was found to be significant for patients without dementia but not for patients with dementia (Figure 3).

## 4. Discussion

This retrospective observational study was conducted to evaluate the factors that affect the discharge ADL of patients who underwent BHA or THA to treat displaced femoral neck fractures. Contrary to the expectations of the surgeons, the choice of surgical method was not statistically significant, at least when the patients were discharged from the hospital. Looking into other factors that may be affected by medical staff, shorter wait times before surgery and longer daily rehabilitation times were associated with improved ADL on discharge (Figure 2). While robust inter-professional collaboration would be necessary to improve these factors, it would be worth the effort to improve the ADL status of patients who underwent surgery for femoral neck fractures. Furthermore, the results of the SASG found that longer daily rehabilitation becomes a significant factor in patients without dementia, making it prudent to prioritize these patients for aggressive rehabilitation considering limited healthcare resources.

Our study has several strengths. First, although MRA often suffers from multicollinearity, we overcame this issue by using SEM, in which significant correlations were considered. This methodology is novel, and we hope to apply this analysis to larger datasets in the future. Second, although this was a single-institution study, it allowed us to suggest specific solutions to the institution. For example, to improve ADL on discharge, the number of days from admission to surgery should be shortened, and the number of rehabilitation units per day should be increased (Table 1, Figure 2). Third, the relationships between variables were visualized by the SEM results, indicating that the direct effect of surgical choice was 2.49, and its indirect effect was −1.46. The results also imply that the choice of THA leads to better ADL and delayed surgery (Figure 2). Finally, to the best of our knowledge, this is the first study to clarify the relationships between multiple factors in femoral neck fractures using SEM.

This study has some limitations. First, this was a single-center study based solely on data derived from NTMC. In Japan, the incidence of hip fractures varies depending on location, with the Kinki, Chugoku, and Kitakyushu regions having high incidence rates [18]. The reasons for these regional differences in incidence have not been clarified, but eating habits (e.g., natto eating habits) and genes (e.g., a theory of old and new mongoloids) are some of the possible factors. Since this study analyzed data from an acute care hospital in Tokyo, it is unclear whether similar results would be found in other regions. Second, this analysis is derived from data during hospitalization only, so the results apply only to the short term and cannot be extrapolated to the long term. Considering that THA is said to be more durable than BHA, it is possible that the long-term effects are better for THA patients. Third, because this study was conducted using information collected from DPC data and electronic medical records, the evidence level of this retrospective observational study was lower than that of a randomized controlled trial. Finally, missing variable bias is possible in MRA. Ideally, we would measure all variables that may affect outcomes and include them in the model; however, since it was not possible to collect additional information from the patients, we cannot deny that a lack of input variables may have led to biases.

According to the survey of hip fractures in Japan, the mean duration of hospitalization was 36.5 days and the mean waiting time before surgery was 4.9 days in 2014 [6], which are in line with other Japanese hospitals but are higher than those of European and American countries. A lack of hip specialists, anesthesiologists, and operating rooms has resulted in longer waiting time for surgery in NTMC and many other hospitals in Japan. Regional cooperative critical pathways have been implemented to shorten hospital stays in acute care hospitals, but the lack of supportive rehabilitation hospitals has hindered their effectiveness to shorten hospital stays.

Since this study included only short-term outcomes, the benefit of longer rehabilitation for patients with dementia was not clear. It is quite possible that longer rehabilitation (e.g., 6 months) could improve ADL and quality of life in these patients. In fact, surgical recommendations may change when high quality studies looking into medium- (e.g., 2 years) and long-term outcomes (e.g., 15 years) have been conducted. Interestingly, Uhler et al. (2017) found that delayed total hip arthroplasty (performed after 48 h) provided greater health utility than early hemiarthroplasty (performed within 48 h), despite the increased 30-day and 1-year mortality associated with delayed surgery [19].

Long-term outcome analysis remains a future goal. This study, using real-world data from an acute care hospital, revealed no significant difference between BHA and THA in terms of ADL on discharge. A multicenter randomized control trial (2019) reported no significant difference between BHA and THA with regard to the risk of unplanned secondary hip procedures over a 24-month period after primary surgery [20], but we hope to investigate factors that may affect long-term ADL outcomes by collecting data from electronic medical records and questionnaires. With increasing pressure to pursue cost-effective treatment plans, we would like to clarify the long-term cost-effectiveness of surgery for displaced femoral neck fractures, possibly by utilizing Markov models using data on the cost and quality-adjusted life years of Japanese people suffering from hip fractures.

## 5. Conclusions

By visualizing multiple factors from real-world data by performing MRA and SEM, our study found that the number of days from admission to surgery and the number of rehabilitation units per day were more important than the surgical selection between BHA and THA when the ADL status at the time of discharge was set as the objective variable. Our results suggest that longer daily rehabilitation is significantly associated with better ADL on discharge, especially in patients without dementia.

## Figures and Tables

**Figure 1 jcm-12-01234-f001:**
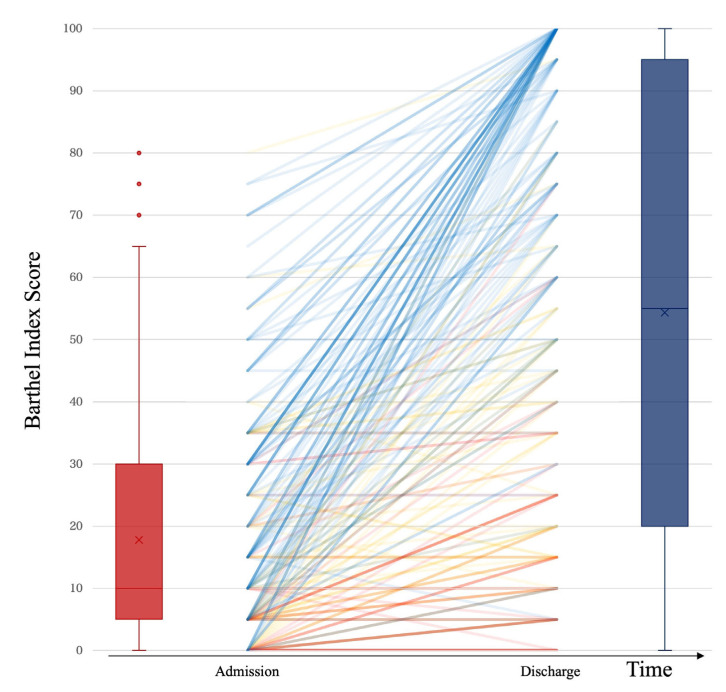
The effect of dementia on ADL changes during hospitalization. ADL is presented using the Barthel Index score (BI: 0–100 points), while the color of the line denotes dementia status: red line, severe dementia (*n* = 47); yellow line, moderate dementia (*n* = 78); and blue line, no cognitive impairment (*n* = 136). Overlapping lines are presented in mixed or overlapping (darker) colors. The red box plot shows BI score on admission, and the blue box plot shows BI score on discharge.

**Figure 2 jcm-12-01234-f002:**
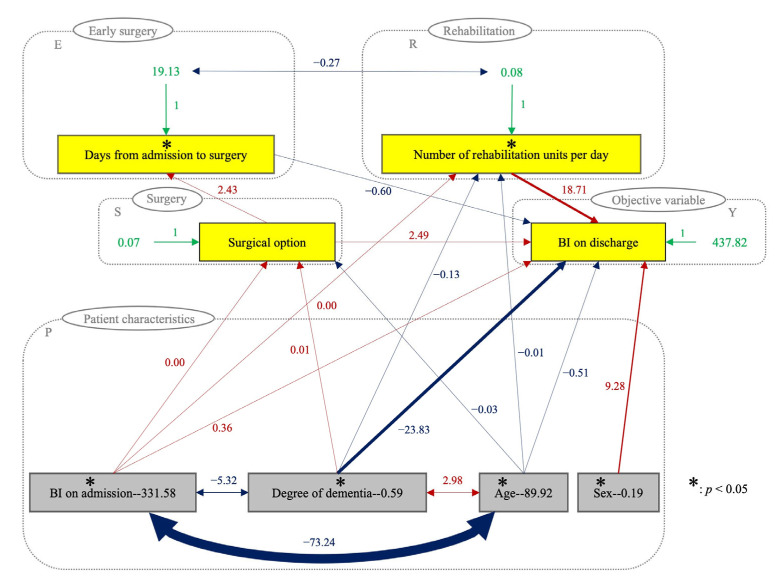
Structural equation modeling with the following conditions: R^2^, 0.65; CFI, 0.99; GFI, 0.98; AGFI, 0.94; and RMSEA, 0.06. The CFI, GFI, AGFI, and RMSEA indicate a good model fit. One-directional arrows indicate causality and two-directional arrows indicate correlation between the variables. Red arrows indicate positive correlation, blue arrows indicate negative correlation, and green arrows indicate the error variance, while line width is proportional to effect. CFI, comparative fit index; GFI, goodness-of-fit index; AGFI, adjusted goodness-of-fit index; RMSEA, root mean square error of approximation.

**Figure 3 jcm-12-01234-f003:**
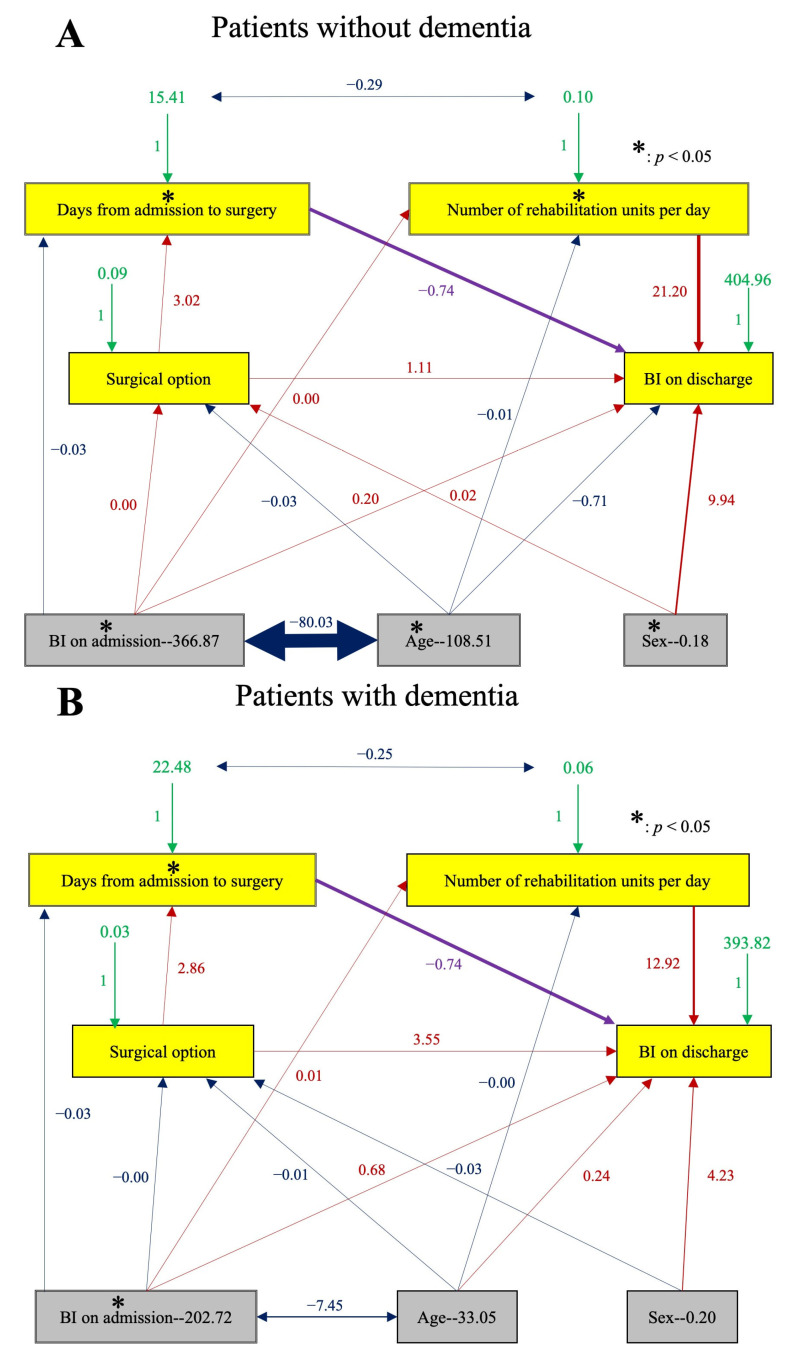
Simultaneous analysis of several groups. CFI, 0.98; GFI, 0.98; AGFI, 0.92; RMSEA, 0.05. The CFI, GFI, AGFI, and RMSEA indicate a good model fit. (**A**) Patients without dementia (*n* = 136). (**B**) Patients with dementia (*n* = 125). Red, positive correlation; blue, negative correlation; green, error variance; purple, equality constraint; line width, proportional to effect. CFI, comparative fit index; GFI, goodness-of-fit index; AGFI, adjusted goodness-of-fit index; RMSEA, root mean square error of approximation.

**Table 1 jcm-12-01234-t001:** Demographic data of BHA/THA patients (*n* = 261).

	BHA (*n* = 223)	THA (*n* = 38)	*p*-Value
Length of hospital stay, mean (SD)	29.5 (16.8)	26.7 (11.1)	0.308
Days from admission to surgery, mean (SD)	4.0 (4.4)	6.9 (4.4)	<0.001
Number of rehabilitation units per day, mean (SD)	1.5 (0.3)	1.6 (0.3)	0.037
Age, mean (SD)	84.5 (6.4)	66.3 (9.8)	<0.001
Female, *n* (%)	170 (76.2)	27 (71.1)	0.541
Patient with dementia, *n* (%)	120 (53.8)	5 (13.2)	<0.001
BI on admission, mean (SD)	15.6 (16.3)	30.5 (23.4)	<0.001

The Student’s *t*-test was used for continuous variables and Fisher’s exact test was used to analyze categorical variables. BHA, bipolar hemiarthroplasty; THA, total hip arthroplasty; SD, standard deviation; BI, Barthel Index.

**Table 2 jcm-12-01234-t002:** Multiple regression analysis.

Explanatory Variables	B ^a^	SE	β ^b^	*p*-Value ^c^	VIF
(Constant)	61.37	25.57	0.00	0.017	
X1: age	−0.51	0.21	−0.14	0.018	2.37
X2: sex	9.28	3.06	0.11	0.003	1.01
X3: degree of dementia	−23.83	2.04	−0.51	<0.001	1.41
X4: BI on admission	0.36	0.08	0.18	<0.001	1.36
X5: days from admission to surgery	−0.60	0.31	−0.08	0.056	1.14
X6: surgical option	2.49	5.24	0.02	0.635	1.98
X7: number of rehabilitation units per day	18.71	4.86	0.17	<0.001	1.49

R^2^, 0.66; adjusted R^2^, 0.65; RMSE, 21.21; *n*, 261; sex: 1, male and 2, female; degree of dementia: 1, no cognitive impairment and 2, moderate dementia and 3, severe dementia; and surgical option: 1, BHA and 2, THA. Abbreviation: SE, standard error; VIF, variance inflation factor; R^2^, coefficient of determination; RMSE, root mean square error; BHA, bipolar hemiarthroplasty; and THA, total hip arthroplasty. ^a^ Unstandardized coefficients, ^b^ standardized coefficients, and ^c^ significant value.

**Table 3 jcm-12-01234-t003:** Correlation and *p*-value.

	X1	X2	X3	X4	X5	X6	X7
X1	1.000 ^†^						
X2	0.038	1.000 ^†^					
X3	0.411 ^†^	0.002	1.000 ^†^				
X4	−0.424 ^†^	−0.026	−0.381 ^†^	1.000 ^†^			
X5	−0.125 *	−0.071	0.050	−0.067	1.000 ^†^		
X6	−0.680 ^†^	−0.043	−0.271 ^†^	0.290 ^†^	0.222 ^†^	1.000 ^†^	
X7	−0.375 ^†^	−0.005	−0.442 ^†^	0.376 ^†^	−0.208 ^†^	0.130 *	1.000 ^†^

* significant difference (*p* < 0.05). ^†^ significant difference (*p* < 0.01). Variables: X1, age; X2, sex; X3, degree of dementia; X4, BI on admission; X5, days from admission to surgery; X6, surgical option; and X7, number of rehabilitation units per day.

**Table 4 jcm-12-01234-t004:** Direct effect, indirect effect, total effect, and *p*-value.

Characteristics	Direct	Indirect	Total	*p*-Value
X1: age	−0.51	−0.15	−0.66	0.013
X2: sex	9.28		9.28	0.002
X3: degree of dementia	−23.8	−2.3	−26.1	<0.001
X4: BI on admission	0.36	0.06	0.42	<0.001
X5: days from admission to surgery	−0.60		−0.60	0.049
X6: surgical option	2.49	−1.46	1.03	0.623
X7: number of rehabilitation units per day	18.7		18.7	<0.001

Direct, direct effect; Indirect, indirect effect; Total, total effect.

## Data Availability

The data will be shared on reasonable request.

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
