# Peer review of "Short-Term Outcomes of Surgery and Rehabilitation on Activities of Daily Living after Displaced Femoral Neck Fractures: Structural Equation Modeling"

_jcm, 2023, doi:10.3390/jcm12031234_

Round 1
Reviewer 1 Report
The Authors should better discuss the causes that determined very high values of " Lenght of hospital stay" and of "Days of waiting before surgery".
The legends to Table 2 and Table 3 should be more clearly rewritten
Reviewer 2 Report
Dear author,
This is a very interesting retrospective observational study with a very good statistical analysis.
From the statistic the ADL at discharge was influence by degrees of dementia and the days of rehabilitation but, most patients with dementia had fewer rehabilitation days. In conclusion longer daily rehabilitation is significantly associated with better ADL on discharge only in patients without dementia. I think that this aspect could be made clearer in the results, and I think that it would be important to know if the ADL improves if you increase more the rehabilitation days for the patients with dementia.
I think you should mention if longer rehabilatation program improves the ADL for patients with dementia.
An important limitation is the very short fallow up (only the hospitalized period), I think that whit longer fallow up the data will change ( the surgical selection), and it is necessary to add this in the discussions and support with literature. The discussion should explain how this limitation influences the data.
